

# 57 years (1960-2017) of snow and meteorological observations from a mid-altitude mountain site (Col de Porte, France, 1325 m alt.)

Yves Lejeune[1], Marie Dumont[1], Jean-Michel Panel[1], Matthieu Lafaysse[1], Philippe Lapalus[1], Erwan Le Gac[1], Bernard Lesaffre[1], and Samuel Morin[1]

[1]Univ. Grenoble Alpes, Université de Toulouse, Météo-France, CNRS, CNRM, Centre d'Etudes de la Neige, Grenoble, France

**Abstract.** In this paper, we introduce and provide access to a daily (1960-2017) and hourly (1993-2017) dataset of snow and meteorological data measured at the Col de Porte site, 1325 m a.s.l, Charteuse, France. Site metadata and ancillary measurements such as soil properties and masks of the incident solar radiation are also provided. Weekly snow profiles are made available from September 1993 to April 2015. A detailed study of the uncertainties originating from both measurements errors and spatial variability within the measurement site is provided for several variables. We show that the estimates of the ratio of diffuse to total shortwave broadband irradiance is affected by an uncertainty of $\pm 0.21$. The estimated root mean squared deviation, that can be mainly attributed to spatial variability, is $\pm 10$ cm for snow depth, $\pm 25$ kg m$^{-2}$ for snow water and $\pm 1$ K for soil temperature ($\pm 0.4$ K during the snow season). The daily dataset can be used to quantify the effect of climate change at this site with a reduction of the mean snow depth (Dec. 1$^{st}$ to April 30$^{th}$) of 39 cm from 1960-1990 to 1990-2017 and an increase in temperature of $+0.90$ K for the same periods. Finally, we show that the daily and hourly datasets are useful and appropriate for driving and evaluating a snowpack model over such a long period. The data are placed on the repository of the Observatoire des Sciences de l'Univers de Grenoble (OSUG) datacenter : http://dx.doi.org/10.17178/CRYOBSCLIM.CDP.2018.

## 1 Introduction

The Col de Porte (CDP) site is a meadow mid-elevation site located at 1325 m altitude (45.30° N, 5.77° E) in the Chartreuse moutain range. This observation site has been operated since 1959 in collaboration with several academic and non-academic partners (https://www.umr-cnrm.fr/spip.php?rubrique218). Daily measurements of snow depth, temperature and precipitation amount have been performed since 1960. Hourly measurement of meterological and snow variables required for running and evaluating detailed snowpack model such as Crocus (Brun et al., 1992; Vionnet et al., 2012) started in 1987 and have been almost continuous during the snow season since snow season 1993-1994. Measured data are manually and automatically checked and corrected using the measurements of several sensors and if required meteorological analyses (SAFRAN, Durand et al., 1999), thus ensuring the quality and continuity of the dataset.

Such dataset provides a unique framework to drive and evaluate snowpack models over long period. Indeed Essery et al. (2013) demonstrated that the evaluation of snowpack models can be misleading if performed over a few snow seasons only. In the recent years, such datasets with a varying level of details have been made public for several snow sites (e.g. Essery et al., 2016) and have motivated the publication of a special issue in Earth System Science Data to gather openly available detailed



meteorological and hydrological observational archives from long-term research catchments in well-instrumented mountain regions around the world. This initiative arises from a GEWEX Hydroclimatology Panel cross-cut project, INARCH, the International Network for Alpine Research Catchment Hydrology.

CDP is part of several observations networks at the local level (Observatoire des Sciences de l'Univers de Grenoble, OSUG),

at the national scale (Observation pour l'Experimentation et la Recherche en Environnement CryObsClim and Systemes d'Observation et d'Experimentation au long terme pour la Recherche en Environnement des glaciers GlacioClim) and contributes to OZCAR (Observatoires de la Zone Critique Applications et Recherches), one of the French components of the eLTER European Research Infrastructure (International Long-term Ecological Research Networks, Gaillardet et al., in review). It is also labeled as a member of the World Meteorological Observation Global Cryospheric Watch Cryonet network

and of the INARCH network. CDP snow and meteorological observations have been selected as an indicator of climate change effect at medium elevation by the national climate change observatory (ONERC). CDP is also an ideal place for specific snow related measurements campaigns, e.g. intercomparison of measurement methods for solid precipitation (SPICE), measurement of the spectral reflectance of snow (Dumont et al., 2017; Tuzet et al., 2017), snow surface roughness (Picard et al., 2016), snow under forest (Sicart et al., 2017).

The objectives of the present paper are (i) extending the hourly dataset published in Morin et al. (2012) from 1993-2011 to 1993-2017, (ii) providing a daily dataset over the 1960-2017 period and (iii) providing estimates of the uncertainties of several variables due to both spatial variability within the observation site and measurements uncertainties. The paper first describes the site and the dataset. The second section is dedicated to provide estimates of measurements uncertainties and spatial variability within the site and the last section describes some examples of the use of this dataset.

Text similar to Morin et al. (2012) is underlined.

## 2 Data description

The Col de Porte site (Fig. 1) is a grassy meadow surrounded by mainly coniferous (spruces) and some lobed-leave trees. All the instruments are located within an area of $40{\times}50$ m$^2$ (Fig. 2, Tabs. 2, 3, 4). The height of the trees ranges from 10 to 40 m. Note that all datasets are provided in universal time coordinate (UTC).

### 2.1 Radiation masks

Surrounding trees and topography are masking part of the shortwave radiation. Masks were measured at location 31 (corresponding to the measurements of the incoming shortwave radiation, see Fig. 2 and Tab. 2) with 5° resolution in azimuth for two dates: July 1998 and June 2018. Masks are provided as a .csv file (doi:10.17187/CRYOBSCLIM.CDP.2018.SolarMask), they contain 3 values for each azimuth that corresponds to: lower elevation, upper elevation and occultation percentage ($p_{occ}$)

defined as follows (Fig. 3). Below the lower mask elevation, there is no direct radiation. Above the upper mask elevation, 100 percent of the direct radiation is available and between the two, only $100 - p_{occ}$ percent of the direct radiation is available. These masks are applied for the calculation of the direct/diffuse shortwave incoming radiation as explained in section 2.3.1.





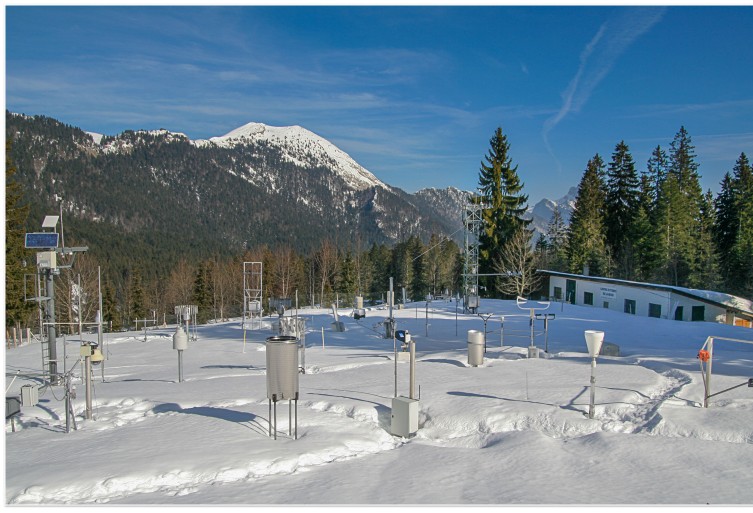

**Figure 1.** Picture of the site taken on 2014-03-10 from the South barrier, looking toward North.

The discrepancies between the two masks are most likely due to changes of the vegetation (growing and major trees cut in 1999, see Morin et al., 2012).

## 2.2   Soil and vegetation properties

Soil properties were measured close to location 33 (see Fig 2) on 29 September 2008, close to location 24 on October 2nd 2012
5   and close to location 30 on 18 October 2017.

On 29 September 2008, the soil properties were measured over the first meter as illustrated by Fig. 4. The layering of the soil was estimated visually and is provided in Tab. 1. The soil properties (particles size analysis, organic matter, nitrogen and carbon total content) were also analyzed down to 87 cm depth. The dataset is provided as a .csv file (soil_properties_2008.csv). On October 2nd 2012, the same analysis was conducted over the first 30 cm of soil at location 24 along with measurements
10   of the soil dry density. The dataset is provided as a .csv file (soil_properties_2012.csv). The two .csv files are available as doi:10.17178/CRYOBSCLIM.CDP.2018.Soil.

**Table 1.** Visual characterization of the soil layers corresponding to Fig. 4 on 29 September 2008.

| Top depth (cm) | Bottom depth (cm) | Visual texture |
|---|---|---|
| 0 | 5 | organic soil with grass roots |
| 5 | 18 | organic soil without root |
| 18 | 47 | clay and sand |
| 47 | 70 | grey clay and sand |
| 70 | 87 | grey clay |
| 87 | 100 | pebbles and grey clay, no sampling |



On 18 october 2017, the soil densities were analyzed for the first 30 cm. At that time, the soil dry density was $1100 \pm 67$ kg m$^{-3}$ without considering the vegetation. The soil wet density was $1475 \pm 59$ kg m$^{-3}$. These values are the mean and standard deviation of 2 measurements over 0-10 cm depth and 2 at 20-30 cm depth close to location 30. No significant differences between the two sampling depths were measured. On the same day, the vegetation (grass) dry and wet mass were measured

5   on a 50 by 50 cm surface at the same location. The measurements result in a value of 1.92 kg m$^{-2}$ for wet mass and 1.54 kg m$^{-2}$ for dry mass. The height of the grass during the time of measurements can be considered as typical for late fall. Note that the grass is frequently cut during summer. These measured soil and vegetation properties can be useful to constrain soil and vegetation schemes which are often coupled with snowpack models (Decharme et al., 2013).



**Figure 2.** Schematic view of the experimental sites with sensor locations. The sensors indicated in yellow are for meteorological variables. The sensors indicated in red are not used anymore as of 2018, those in blue corresponds to snow measurements. Areas 23 and 24 corresponds to soil temperature and humidity measurements. The correspondance between numbering and sensors are indicated in Tabs. 2, 3 and 4. The 3 emoticons correspond to the 3 Webcam locations.

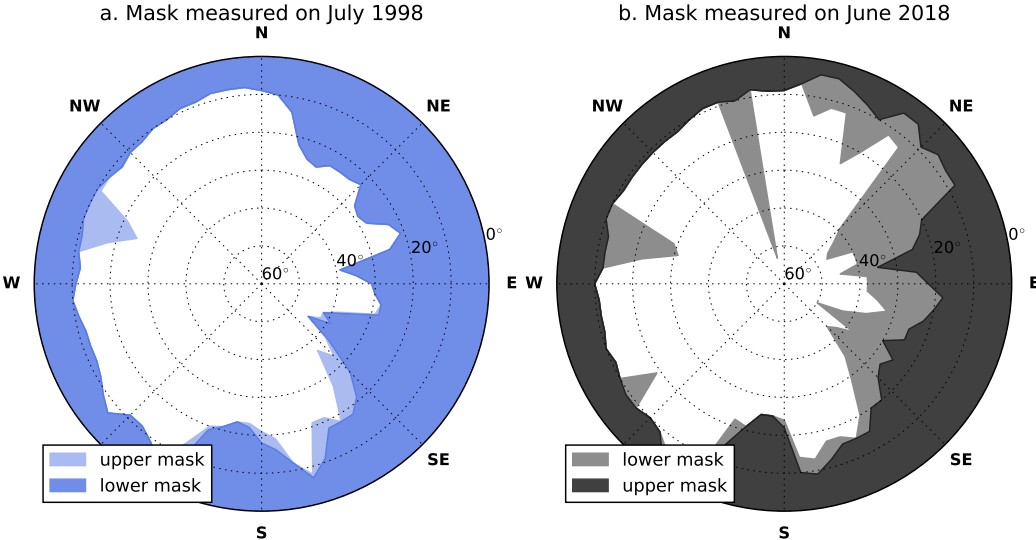

**Figure 3.** Masks measured at location 31 on July 1998 (left panel) and on June 2018 (right panel). Upper and lower mask elevations are represented by the solid lines. Elevations are given in degrees, the center is 60 degrees elevation.



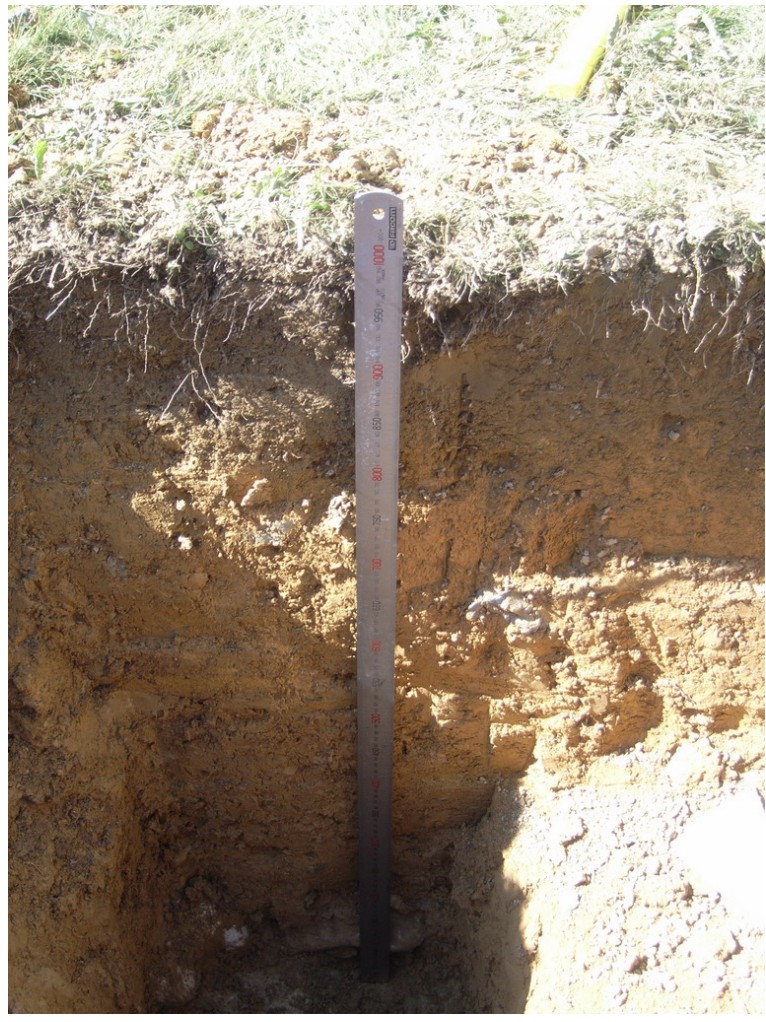

**Figure 4.** Soil profile of 1 meter depth performed close to location 33 on 29 September 2008.



## 2.3 Meteorological hourly data, 1993-2017

The meteorological hourly dataset over 1993-2017 is an extension of the meteorological dataset provided in Morin et al. (2012). An extensive description of the dataset is available in this study. Below are reported only changes that happened after 2011 and additional details not provided in Morin et al. (2012).

The dataset is provided as a continuous hourly dataset since 1993, so that it can be easily used to drive snowpack models. The partitioning of the dataset between *in-situ* data and the output of the meteorological analysis and downscaling tool SAFRAN (Durand et al., 1999) is the same as in figure 4 of Morin et al. (2012). For years 2011 to 2015, *in-situ* data are restricted to the period 20th October of one year to 10th June of the next year. Starting on 10th June 2015, all data are *in-situ* year-round except for very short periods with observation issues. An *in situ* flag is provided together with the meteorological data (value = 1 for

*in situ* data).

Table 2 provides an udpate of the type of sensors used for meteorological measurements with respect to Tab. 1 in Morin et al. (2012). The dataset is provided in netCDF format (doi:10.17178/CRYOBSCLIM.CDP.2018.MetInsitu) in the standard format for SURFEX surface model meteorological inputs (Vionnet et al., 2012; Masson et al., 2013). The atmopsheric pressure value corresponds to the mean climatological value at CDP.

### 15   2.3.1 Shortwave incoming radiation

The meterological dataset provides both total and diffuse incoming broadband radiations at location 31. The diffuse shortwave radiation is not measured but calculated from total shortwave and longwave incident radiation and air temperature as described in the following.

The first step of the procedure is to compute a cloudiness value, $\eta$ (no unit, between 0 for clear sky and 1 for fully overcast)

from measured air temperature $T_{air}$ (K), longwave radiation $LW_{down}$ (W m$^{-2}$) and specific humidity using Equations 1 and 2 from Berland (1952).

$$LW_{down} = \varepsilon \sigma T_{air}^4 \tag{1}$$

$$\varepsilon = 0.58 + 0.9(0.09 + 0.2\eta)\eta^2 + 0.06e_{air} - 0.05 * (1 - (0.09 + 0.2\eta)\eta^2) \tag{2}$$

where $\sigma$ is the Stefan-Boltzman constant, and $e_{air}$ is the water vapour partial pressure calculated from measured $T_{air}$ and

relative humidity, expressed in hPa.

The calculated value of $\eta$ is then used to partition the total measured shortwave radiation into direct and diffuse fraction using the radiative transfer model from Vauge (1983) and the measured mask described in Sec. 2.

An additional shortwave radiation sensor (Delta-T SPN1 -heated) has been installed at location 5 (Fig. 2) in September 2016 (9.5 m above ground) and measures both diffuse and total shortwave radiation over 400-2700 nm range.

A comparison between these measured and calculated direct/diffuse distribution is provided in Sec. 3.1.



### 2.3.2  Longwave incident radiation

The sensor for incident longwave radiation was replaced in October 2015 by a Kipp&Zonen CGR4 sensor (location 30 in Fig. 2). Figure 5 displays the comparison of the measured incident longwave radiation with simulated longwave radiation from SAFRAN based on monthly average. It shows that the bias between SAFRAN and the measurements displays two large

5    breaks of 10 W m$^{-2}$ in October 2015 and in autumn 2010 (corresponding to another sensor replacement, Tab. 2). Based on the hypothesis that the newest sensor is the reference since it was fully calibrated oustide and inside with a blackbody, the dataset was corrected as follow : +10 W m$^{-2}$ from 1993 to November 2010 and -10 W m$^{-2}$ from November 2010 to November 2015. This correction, although ranging in the uncertainty values provided by sensor manufacturor, is of large significance for snowpack modelling considering the high sensitivity of the snowpack to this variable (e.g. Raleigh et al., 2015; Sauter and

10   Obleitner, 2015; Quéno et al., 2017).

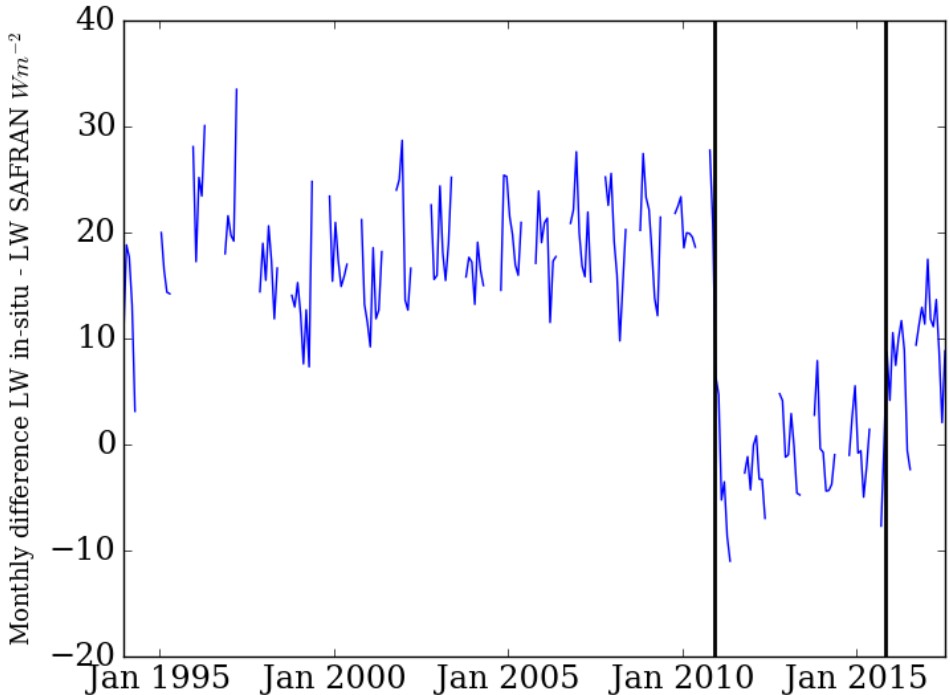

**Figure 5.** Monthly average of the difference between measured downward longwave and SAFRAN estimates. The two vertical black lines indicate the sensor changes (cf Tab. 2).



**Table 2.** Overview of the sensors used to gather the hourly meteorological data, between 1993 and 2017 at Col de Porte, France. The index refers to the location of the sensor represented in Fig. 2. Text similar to Morin et al. (2012) is underlined.

| Variable | Index | Sensor | Period of operation | Height | Unit | Integration method |
|---|---|---|---|---|---|---|
| Air temperature | 12 | PT 100/ 3 wires | ... → 1996/1997 | 1.5 m* | K | Instantaneous |
| | 12 | PT 100/ 4 wires | 1997/1998 → ... | 1.5 m* | K | Instantaneous |
| | mast | PT 100/ 4 wires | 1997/1998 → ... | 3.1 m | K | Instantaneous |
| Relative humidity | 13 | SPSI MU-C.1/MUTA.2 | ... → 1994/1995 | 1.5 m* | %RH | Instantaneous |
| | 13 | Vaisala HMP 35DE | 1995/1996 → 2005/2006 | 1.5 m* | %RH | Instantaneous |
| | 13 | Vaisala HMP 45D | 2006/2007 → ... | 1.5 m* | %RH | Instantaneous |
| Windspeed | 2 | Laumonier – heated | 1997/1998 → ... | 10 m | $\mathrm{m\,s^{-1}}$ | Integrated (60 min) |
| | 7 | Chauvin Arnoux Tavid 87 – non-heated | whole record | 10 m | $\mathrm{m\,s^{-1}}$ | Integrated (60 min) |
| | 15 | Laumonier – heated | 2000/2001 → 2014-2015 | 3.3 m | $\mathrm{m\,s^{-1}}$ | Integrated (60 min) |
| | 3 | Thies Ultrasonic anemometer – heated | March 2012 → ... | 10 m | $\mathrm{m\,s^{-1}}$ | Integrated (60 min) |
| | 18 | Thies Ultrasonic anemometer – heated | Dec. 2013 → ... | 3,3 m | $\mathrm{m\,s^{-1}}$ | Integrated (60 min) |
| Inc. shortwave radiation | 31 | Kipp & Zonen CM7 | ... → 15 March 1996 | 1.2 m* | $\mathrm{W\,m^{-2}}$ | Integrated (50 min) |
| | 31 | Kipp & Zonen CM14 | 15 March 1996 → Oct. 31st 2015 | 1.2 m* | $\mathrm{W\,m^{-2}}$ | Integrated (50 min) |
| | 31 | Kipp & Zonen CMP10 | Nov. 2015 → ... | 1.2 m* | $\mathrm{W\,m^{-2}}$ | Integrated (50 min) |
| Inc. longwave radiation | 30 | Eppley PIR | ... → 2010/2011 | 1.2 m* | $\mathrm{W\,m^{-2}}$ | Integrated (50 min) |
| | 30 | Kipp & Zonen CG4 | 2010/2011 → Oct. 2015 | 1.2 m* | $\mathrm{W\,m^{-2}}$ | Integrated (50 min) |
| | 30 | Kipp & Zonen CRG4 | Oct. 2015 → ... | 1.2 m* | $\mathrm{W\,m^{-2}}$ | Integrated (50 min) |
| Precipitation | 9 | PG2000 heated (2000 cm²), tipping bucket | whole record | 2.75 m | $\mathrm{kg\,m^{-2}\,s^{-1}}$ | Difference |
| | 1 | PG2000 non-heated (2000 cm²), tipping bucket | whole record | 2.75 m | $\mathrm{kg\,m^{-2}\,s^{-1}}$ | Difference |
| | 20 | GEONOR (200 cm²) with windshield, weighing gauge | whole record | 3 m | $\mathrm{kg\,m^{-2}\,s^{-1}}$ | Difference |
| | 17° | GEONOR T-200B-3 (200 cm²), weighing gauge | Dec. 2013 → ... | 3.1 m | $\mathrm{kg\,m^{-2}\,s^{-1}}$ | Difference |
| | 19° | GEONOR T-200B-3 (200 cm²) with windshield, weighing gauge | Dec. 2013 → ... | 3.1 m | $\mathrm{kg\,m^{-2}\,s^{-1}}$ | Difference |
| | 34° | OTT Pluvio 2 OTT (400 cm²) with windshield, weighing gauge | Dec. 2013 → ... | 3.1 m | $\mathrm{kg\,m^{-2}\,s^{-1}}$ | Difference (amount NRT) |

\* Height adjusted manually above snow surface (≈ weekly).

° The sensors have been installed for the WMO SPICE project and are used in this study only to complement the dataset if a problem exists for the reference sensor.

### 2.3.3 Precipitation

Precipitation data are handled according to Morin et al. (2012). Precipitation data are manually partitioned between liquid and solid phase using all relevant sources of data at the site. The precipitation values provided in the dataset are based on the reference gauge, GEONOR, at location 20. Other OTT and GEONOR gauges are used in complement to the reference
5  sensor measurements. Hourly solid precipitation measurements are corrected for undercatch depending on temperature and wind speed, as described in Morin et al. (2012). From 2013 to 2017, the wind measurement used for the correction is placed at location 18 (Fig. 2), which better represents the wind at the level of the different precipitation gauges.





## 2.4 Snow and soil data, 1993-2017

The evaluation hourly dataset over 1993-2017 is an extension of the evaluation dataset provided in Morin et al. (2012). An extensive description of the dataset is available in the latter study. Below are reported only changes that happened after 2011 and additional details not provided in Morin et al. (2012). The hourly dataset is provided as a netCDF file

(doi:10.17178/CRYOBSCLIM.CDP.2018.HourlySnow). Within this dataset, the soil temperature, soil humidity and settling disk temperature are raw measurements (uncorrected).

Table 3 provides an udpate of the type of sensors used for evaluation measurements with respect to Tab. 2 in Morin et al. (2012).

Starting on October 2010, the snow depth at location 32 has been measured with a Dimetix Laser ranger. The field-of-view

is a few mm diameter spot and the accuracy provided by the manufacturer is $\pm$ 1.5 mm. Since October 2010, the snow depth measurements provided in the dataset (reference snow depth) is the measurements of the Dimetix Laser ranger. Data from the other sensors are used to evaluate and correct the Laser data.

The surface temperature reference values contained in the dataset mainly originates from the Kipp&Zonen updward pyrgeometer (location 25, same sensor as location 30 in Tab. 2). Since september 2010, these data are complemented by the other

surface temperature sensors with a conical field of view shown in Tab. 3. The reference surface temperature is bounded to 273.15 K when snow is present on the ground.

New sensors for soil temperature and humidity have been installed in October 2012 at several depths (-0.05,-0.1,-0.2,-0.3 m) at location 23 close (roughly 2 m) to location 24 (Fig. 2) where the older soil temperature sensors were located. The differences between the measurements at these two locations are discussed in Sec. 3.4. It must be underlined that the soil

humidity measurements show that the soil is almost always saturated by liquid water when snow is present. This characteristic may not be typical for mountain slopes (e.g. Williams et al., 2009) and may be difficult to reproduce with usual soil models.

The measurements of the vertical profile of snowpack properties as described in Fierz et al. (2009) are also provided in caaml format (version 5) according to the International Association for Cryospheric Science (IACS) standard. They can be visualized using the niViz software. An example is displayed in Fig. 6 for 13 January 2001. These profiles are available on a weekly basis

from September 1993 to April 2015 (doi:10.17178/CRYOBSCLIM.CDP.2018.SnowProfile).





**Table 3.** Overview of the sensors used to gather the hourly and daily snow and soil data, between 1993 and 2017 at Col de Porte, France. Note that outgoing shortwave and longwave radiation is measured using instruments similar to the corresponding incoming radiation, described in Table 2. Note also that snow surface temperature can be derived from the outgoing longwave radiation sensor, in addition to the sensors presented here. The index refers to the location of the sensor represented in Fig. 2. Text similar to Morin et al. (2012) is underlined.

| Variable | Index | Sensor | Period of operation | Height | Unit | Time resolution | Integration method |
|---|---|---|---|---|---|---|---|
| Snow depth | 33 | BEN ultrasonic depth gauge | ... → 1999/2000 | 3 m | m | hourly | Instantaneous |
| | 33 | FNX ultrasonic depth gauge | 2000/2001 → 2008/2009 | 3 m | m | hourly | Instantaneous |
| | 33 | Campbell Ultra-sound depth gauge SR50A | 2009/2010 → ... | 3.5 m | m | hourly | Instantaneous |
| | 32 | Dimetix Laser ranger | 2010/2011 → ... | 3.1 m | m | hourly | Instantaneous |
| | 6° | Campbell Ultra-sound depth gauge SR50 | Jan. 2014 → ... | 4.1 m | m | hourly | Instantaneous |
| | 6° | Campbell Ultra-sound depth gauge SR50ATH | Jan. 2014 → ... | 4.1 m | m | hourly | Instantaneous |
| | 6° | Jenoptik Laser ranger | Jan. 2014 → ... | 4.1 m | m | hourly | Instantaneous |
| | 6° | Dimetix Laser ranger | Jan. 2014 → ... | 4.1 m | m | hourly | Instantaneous |
| | hatched | Snowpit (up to three values) | whole record | N.A. | m | ≈ weekly | N.A. |
| Snow water equivalent | 16 | Cosmic-Ray Neutron sensor | 2001/2002 → ... | 0 m | $kg\,m^{-2}$ | daily | 24h integration |
| | 16 | Cosmic-Ray Neutron sensor[a] | 2008/2009 → ... | 0 m | $kg\,m^{-2}$ | daily | 24h integration |
| | hatched | Snowpit (up to three values) | whole record | N.A. | $kg\,m^{-2}$ | ≈ weekly | N.A. |
| Runoff | 11 | 5 m² lysimeter – tipping gauge | ... → March 1994 | 0 m | $kg\,m^{-2}\,s^{-1}$ | hourly | Difference |
| | 11 | 5 m² lysimeter – scale | March 1994 → ... | 0 m | $kg\,m^{-2}\,s^{-1}$ | hourly | Difference |
| | 14 | 1 m² lysimeter – tipping gauge | ... → Dec. 1996 | 0 m | $kg\,m^{-2}\,s^{-1}$ | hourly | Difference |
| | 14 | 1 m² lysimeter – scale | Dec. 1996 → ... | 0 m | $kg\,m^{-2}\,s^{-1}$ | hourly | Difference |
| Surface temperature | 22 | Testo term Pyroterm | ... → 2016/10 | 1.2 m[b] | K | hourly | Instantaneous |
| | 21 | Campbell IR120 | Nov. 2015 → ... | 0.8 m[b] | K | hourly | Instantaneous |
| | 28 | Heitronics KT15 | 2010/2011 → ... | 3.2 m | K | hourly | Instantaneous |
| | 4° | Campbell IR120 | Jan. 2014 → ... | 4.1 m | K | hourly | Instantaneous |
| Soil temperature | 24 | PT 100/3 wires | ... → 1996/1997 | −0.1 m | K | hourly | Instantaneous |
| | 24 | PT 100/ 4 wires | 1997/1998 → ... | | | | |
| | 24 | PT 100/ 3 wires | ... → 1996/1997 | −0.2 m | K | hourly | Instantaneous |
| | 24 | PT 100/ 4 wires | 1997/1998 → ... | | | | |
| | 24 | PT 100/ 3 wires | ... → 1996/1997 | −0.5 m | K | hourly | Instantaneous |
| | 24 | PT 100/ 4 wires | 1997/1998 → ... | | | | |
| | 23 | PT 100/ 4 wires | Oct. 2012 → ... | −0.05 m | K | hourly | Instantaneous |
| | 23 | PT 100/ 4 wires | Oct. 2012 → ... | −0.10 m | K | hourly | Instantaneous |
| | 23 | PT 100/ 4 wires | Oct. 2012 → ... | −0.20 m | K | hourly | Instantaneous |
| | 23 | PT 100/ 4 wires | Oct. 2012 → ... | −0.30 m | K | hourly | Instantaneous |
| Soil moisture | 23 | Delta-T ML2x ThetaProbe Moisture sensor | 2012/10 → ... | −0.05 m | $m^3\,m^{-3}$ | hourly | Instantaneous |
| | 23 | Delta-T ML2x ThetaProbe Moisture sensor | Oct. 2012 → ... | −0.10 m | $m^3\,m^{-3}$ | hourly | Instantaneous |
| | 23 | Delta-T ML2x ThetaProbe Moisture sensor | Oct. 2012 → ... | −0.20 m | $m^3\,m^{-3}$ | hourly | Instantaneous |
| | 23 | Delta-T ML2x ThetaProbe Moisture sensor | Oct. 2012 → ... | −0.30 m | $m^3\,m^{-3}$ | hourly | Instantaneous |
| Settling disks temp. | 27 and 29 | PT 100/3 wires | ... → 1996/1997 | variable | K | hourly | Instantaneous |
| | 27 and 29 | PT 100/ 4 wires | 1997/1998 → ... | | | | |
| Settling disks height | 27 and 29 | In-house positioning system | whole record[c] | variable | m | hourly | Instantaneous |
| Ground flux | 24 | Hukseflux HFP01 | since 2010/2011 | 0 | $W\,m^{-2}$ | hourly | Instantaneous |

[a] Sensor including a shielding for ground-originating neutrons (reduced data scatter).

[b] Height adjusted manually above snow surface (≈ weekly).

[c] Progressive migration from mercury to solid state electric contact.

° The sensors have been installed for the WMO SPICE project and are used in this study only to complement the dataset if a problem exists for the reference sensor.



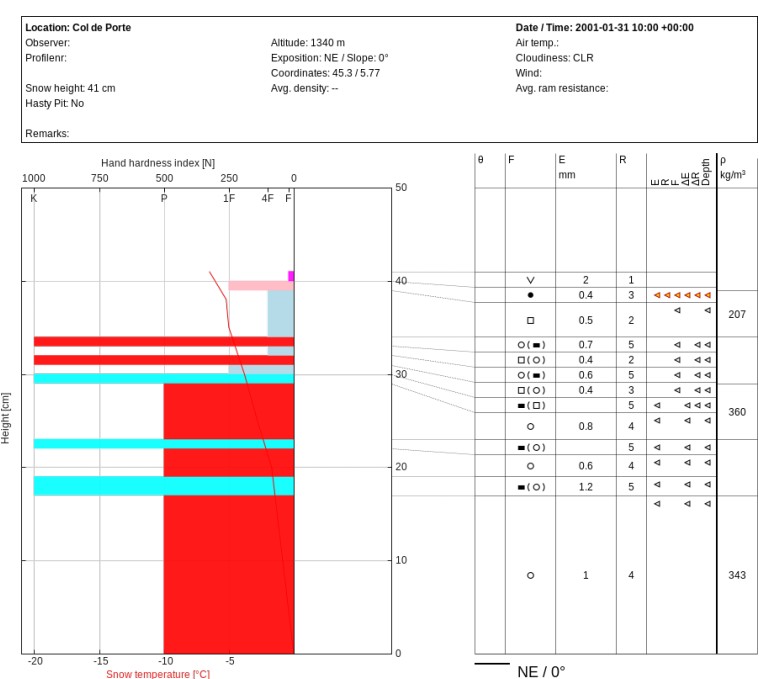

**Figure 6.** Example of snow profile measured on 13 January 2001 vizualized using niViz software.





### 2.5 1960-2017 Data

Table 4 describes the daily dataset that combines snow and meteorological measurements. The dataset is provided in netCDF format (doi:10.17178/CRYOBSCLIM.CDP.2018.MetSnowDaily). Variable names correspond to the names listed in Tab. 4. Within this daily dataset, the total precipitation dataset is not corrected for undercatch contrary to rain and snow datasets

5 (starting in Sept. 1993). The total precipitation dataset is also not measured by the same sensor as the rain and snow datasets (cf Tab. 4). The total precipitation dataset is measured with the PG2000 sensor thus mitigating the impact of the undercatch correction. In addition, the total precipitation time serie may be qualified as inhomogeneous in time due to the various changes in precipitation gauges. The daily SWE automatic measurements (loc. 16) are discarded for snow season 2015/2016 due to a disfunction of the sensor. Note also that the daily albedo data are uncorrected for local snow surface slope.

10 The hourly meteorological dataset that contains the whole SAFRAN reanalysis (Durand et al., 2009) at Col de Porte for the period 1960-2017 is provided in order to drive snowpack simulation over the whole period. The dataset is provided in netCDF format (doi:10.17178/CRYOBSCLIM.CDP.2018.MetSafran) in the standard format for SURFEX meteorological inputs (Vionnet et al., 2012; Masson et al., 2013). The solar mask measured in 1998 (Fig. 3) is accounted for in this dataset.





**Table 4.** Description of the daily dataset between 1960 and 2017 at Col de Porte, France. The index refers to the location of the sensor represented in Fig. 2.

| Variable | Index | Sensor | Period of operation | Unit | Description |
|---|---|---|---|---|---|
| $T_{min}$ | 12 | PT100 | ... → 1993 | K | Min. temp. between 00:00 (day D) and 24:00 (day D) |
| | 12 | cf Tab. 2 | 1993 → ... | K | Min. temp between 06:00 (day D-1) and 06:00 (day D) |
| $T_{max}$ | 12 | PT100 | ... → 1993 | K | Max. temp. between 00:00 (day D) and 24:00 (day D) |
| | 12 | cf Tab. 2 | 1993 → ... | K | Max. temp between 06:00 (day D) and 06:00 (day D+1) |
| snow_depth_auto | close to 33 | automatic sensor | ... → 1977/1978 | m | Snow depth 06:00 (day D) |
| | 33 | Ultra-sound depth gauge BEN | 1978/1979 → 1999/2000 | m | Snow depth 06:00 (day D) |
| | 33-6 | cf Tab. 3 | 1993 → ... | m | Snow depth 06:00 (day D) |
| snow_depth_pit | hatched | manual | 1963/1964 → 7 Feb. 1996 | m | Irregular frequency |
| | hatched | manual | 8 Feb. 1996 → ... | m | Weekly |
| snow_depth_pit_north | hatched | manual | 2001/2002 → ... | m | Weekly |
| snow_depth_pit_south | hatched | manual | 2001/2002 → ... | m | Weekly |
| swe_auto | 16 | cf Tab. 3 | 2001/2002 → ... | kg m$^{-2}$ | Daily (not available for 2015-2016) |
| swe_pit | hatched | manual | 1963/1964 → 7 Feb. 1996 | kg m$^{-2}$ | Irregular frequency, SWE core 38.5 and 25. cm$^2$ |
| | hatched | manual | 8 Feb. 1996 → ... | kg m$^{-2}$ | Weekly, SWE core 100 cm$^2$ |
| swe_pit_north | hatched | manual | 2001/2002 → ... | kg m$^{-2}$ | Weekly, SWE core 100 cm$^2$ |
| swe_pit_south | hatched | manual | 2001/2002 → ... | kg m$^{-2}$ | Weekly, SWE core 100 cm$^2$ |
| total_precipitation | 9 | cf Tab. 2 | 1960/1961 → 2004/2005 | kg m$^{-2}$ | Daily sum of precipitation not corrected for undercatch 06:00 (day D) to 06:00 (day D+1) |
| rain$^\diamond$ | 20 | cf Tab. 2 | 1993/1994 → ... | kg m$^{-2}$ | Daily sum of corrected liquid precipitation, 06:00 (day D) to 06:00 (day D+1) |
| snow$^\diamond$ | 20 | cf Tab. 2 | 1993/1994 → ... | kg m$^{-2}$ | Daily sum of corrected solid precipitation, 06:00 (day D) to 06:00 (day D+1) |
| height of new snow | 33,27 | calculated from snow depth measurement and settlement disks | whole record | cm | Daily sum of new snow, 06:00 (day D) to 06:00 (day D+1) |
| albedo_daily | 26 and 31 | cf Tab. 2 | 2005/2006 → ... | NA | Ratio of the daily sums of reflected and incident shortwave radiations |
| albedo_daily_flag | 26 and 31 | NA | 2005/2006 → ... | NA | Number of hourly measurements used to calculate daily albedo |

$^\diamond$ Note that rain and snow variables are provided only when *in situ* measurements are available (i.e *in situ* flag of Tab. 2 - see also figure 4 in Morin et al., 2012).





## 3 Spatial variability and measurements uncertainties

The dataset presented in this study is, like any observation dataset, affected by different sources of uncertainties. Regardless whether these data should be used for model evaluation or process study, characterizing their associated uncertainties is essential for a proper use of the data. The uncertainties of the dataset may come from measurements uncertainties (including

instrumental and environmental uncertainties) but also from the spatial variability of the variables within the measurement plot.

A lower bound of the uncertainty for each variable can be estimated from the information provided by the sensor manufacturer. Some variables are measured at different locations within the field sites and by different sensors. This provides a better insight of the uncertainty associated to both sources for each variable. Lafaysse et al. (2017) already provided a first estimate

of the uncertainty associated to snow depth, snow water equivalent, bulk density, broadband albedo, soil temperature and snow surface temperature. In this section, we extend the period and the number of points used for the uncertainties evaluation for snow depth, snow water equivalent and soil temperature for which several measurements are available over a sufficiently long period. We also provide uncertainties assessment of the direct/diffuse incident shortwave radiation estimates (cf sec. 2.3.1). Note that an update on the uncertainties for snow surface temperature and broadband albedo is not provided in this study (lack

of a sufficient number of sensors) though their uncertainty estimates are crucial for snow model evaluation. In this respect, we recommend the use of uncertainty values provided in Lafaysse et al. (2017) for these two variables.

### 3.1 Direct/diffuse shortwave incoming radiation

A first source of uncertainties in the calculation of the distribution of the measured broadband shortwave radiation into diffuse and direct radiation originates from the uncertainties of the mask used for the calculation (cf sec. 2, Fig. 3). Using the

methodology explained in Sec. 2.3.1, we estimate the direct and diffuse shortwave incoming radiation based on the mask from 1998 and the mask from 2018 for two snow seasons (Sept. 1$^{\text{st}}$ to 30 June ) : 2015-2016 and 2016-2017. The mean difference (mask measured in 2017 minus mask measured in 1998) and root mean square deviation (RMSD) computed between diffuse components (over non zero values only) are -1.30 W m$^{-2}$ and 10.1 W m$^{-2}$. The mean difference and RMSD for the diffuse to total ratio are -0.02 and 0.10. The histogram of differences are provided in Fig. 7a.

The accuracy of the methodology described in Sec. 2.3.1 has also been evaluated using the measurements of total and diffuse radiations from location 2 (at 10 m above ground) and the mask measured in October 2017 at the same location. The comparison is done from September 1$^{\text{st}}$ 2016 to 30 June 2017 during daylight (i.e., if total measured shortwave larger than 4 W m$^{-2}$ ). The mean difference between the estimated and simulated diffuse component is -15.26 W m$^{-2}$ (RMSD: 53 W m$^{-2}$). The mean difference and RMSD computed for the diffuse to total ratio are -0.08 and 0.21. The histograms of differences

are provided in Fig. 7b. This shows that the estimation of the diffuse radiation is slightly biased low and that this uncertainty has to be taken into account for applications such as radiative balance calculation for which the direct/diffuse distribution has a significant impact. It also shows that the methodology applied to partition the direct and diffuse components has a larger impact on the uncertainty than the change in solar masks shown in Fig. 3.




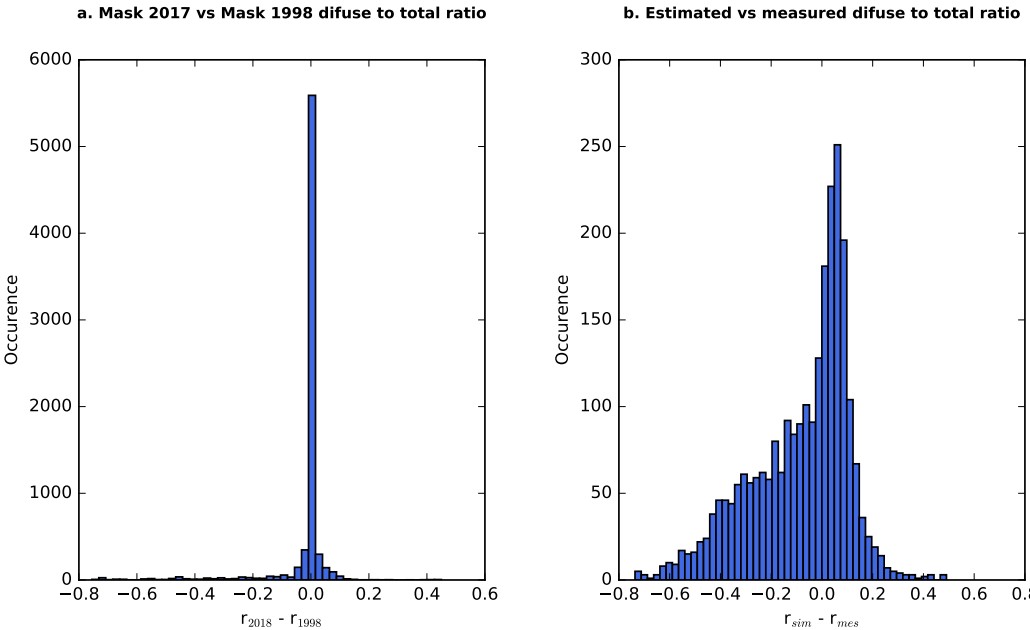

**Figure 7.** Comparison of different broadband diffuse to total shortwave radiation ratio, $r$. (a) Difference in ratio estimated with the mask measured in June 2018 and in June 1998 at location 25. Statistics are calculated during daylight from Sep. 1$^{st}$ 2015 to 30 June 2017 excluding July and August for each year. (b) Difference in ratio estimated with the 2017 mask (measured at location 5, October 21st 2017) and the measured ratio at location 5. Statistics are calculated during daylight from September 1$^{st}$ 2016 to 30 June 2017.

**Table 5.** Statistics of the comparisons between the different snow depth measurements represented in Fig. 8.

| Sensors | Number of dates | Deviation (m) | RMSD (m) | Period |
|---------|-----------------|---------------|----------|--------|
| Nivose 1 - $h_{ref}$ | 22498 | -0.007 | 0.039 | Sept. 2009 to June 2016 |
| Mast - $h_{ref}$ | 22225 | 0.013 | 0.036 | Sept. 2009 to June 2016 |
| Pit - $h_{ref}$ | 895 | 0.054 | 0.078 | Sept. 1960- June 2017 |
| North Pit - $h_{ref}$ | 233 | 0.133 | 0.136 | Sept. 2001 to June 2017 |
| South Pit - $h_{ref}$ | 233 | 0.109 | 0.111 | Sept. 2001 to June 2017 |

## 3.2 Snow depth

Figure 8 compares the snow depth reference value mostly measured at location 32-33, $h_{ref}$ (Fig. 2) with several other measurements of snow depth : in panel (a) with respect to automatic snowdepth measurements at locations "Nivose 1" and 6 (see Fig. 2) and in panel (b) with respect to manual snow depth measurement in snow pit fields (main, north and south, blue hatched areas in Fig. 2). For panel (a), the comparison is done over the 2009-2016 period and the automated measurement have been manually corrected for any blank period and measurements inconsistency. For panel (b), the comparison with the main snow





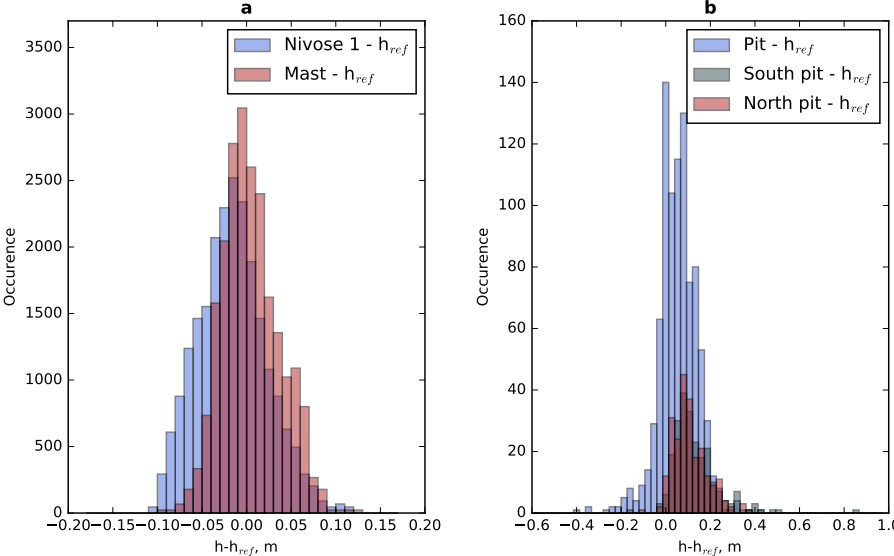

**Figure 8.** Comparison of snow depth measurements at different locations. $h_{ref}$ corresponds to location 33. (a) Difference in measured snow depth between the ultrasound sensor placed on the Nivose 1 location (Fig. 2) and the reference snow depth (locations 32-33 in Fig. 2) in blue. In red, differences between the measured snow depth at location 6 (Fig. 2) and the reference snow depth. The difference are calculated from snow season 2009/2010 to snow season 2015/2016 using only data from 20 Sept. to 10 June . Data where both locations indicate 0 snow depth are excluded from the statistics. (b) Difference in measured snow depth between the manual snow depth measurements at snowpit field location (Fig. 2) and reference automatic snow depth (location 33) in blue, between manual snow depth measurement in the snow pit south field and reference in grey and snow pit north field and reference in red. Difference values are calculated over 1960-2017 period for the pit value and 2001-2017 for north and south pits. Data where both locations indicate 0 snow depth are excluded from the statistics. Corresponding statistics are provided in Tab. 5.

pit field is done over 1960-2017 and for the pits south and north over 2001-2017. For each sensor, the number of points used to calculate the statistics are in Tab. 5.

Figure 8a and Table 5 show that the three automatic measurements exhibit deviations lower than 1.3 cm and that the RMSD is lower than 4 cm. Higher discrepancies are found between the reference automatic measurements and the manual measurements (Figure 8b) with mean deviation reaching almost 14 cm and RMSD 14 cm. These higher difference values might be attributed to the local slope, aspect, and small topographic features within the three snow pit fields area and to the higher measurements uncertainties associated to manual measurements. Picard et al. (2016) installed during the 2014-2015 snow season an automatic scanning laser meter close to location 6 that scanned an area of 100-200 m$^2$. During this snow season, the laser measurements indicated a spatial varibility of the snow depth within the footprint that can reach 7-10 cm (RMSD). We thus recommend the





use of $\pm$ 10 cm uncertainty value for snow depth in any evaluation to represent the spatial variability within the site, that is comparable to the values used in Lafaysse et al. (2017).

## 3.3 Snow water equivalent

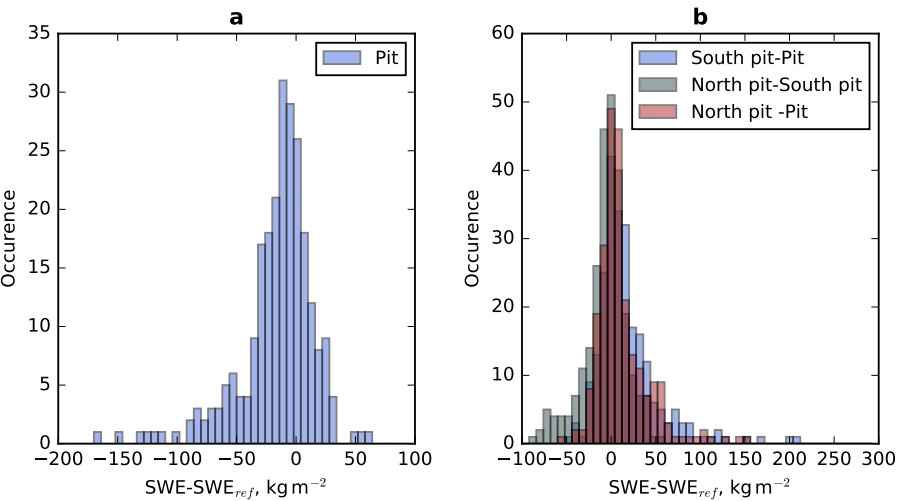

**Figure 9.** Comparison of SWE measurements at different locations. (a) Difference in measured SWE between the manual measurement in the snow pit field (Fig. 2) and the reference automatic SWE ($SWE_{ref}$, location 16 in Fig. 2) in blue. The difference are calculated over the period 2001-2017 (no reference data for 2015/2016 snow season). Data where both locations indicate 0 SWE are excluded from the statistics. Note the manual measurements from snow pit south and north are used for the SWE sensor (location 16) calibration. (b) Difference in manually measured SWE between the snowpit field south and the snow pit field location in blue, between the snow pit field north and south locations in green and snow pit north field and snow pit field in red. Difference are calculated over 2001-2017 period. Data where both locations indicate 0 SWE are excluded from the statistics. Numerical values are provided in Tab. 6.

**Table 6.** Statistics of the comparisons between the different SWE measurements represented in Fig. 9.

| Sensors | Number of dates | Deviation ($kg\,m^{-2}$) | RMSD ($kg\,m^{-2}$) | Period |
|---|---|---|---|---|
| Pit - $SWE_{ref}$ | 244 | -16.83 | 24.44 | Sept. 2001 to June 2017 |
| South Pit - Pit | 239 | 17.37 | 25.09 | Sept. 2001 to June 2017 |
| North Pit - South Pit | 260 | -6.69 | 17.66 | Sept. 2001 to June 2017 |
| South Pit - Pit | 239 | 11.84 | 20.01 | Sept. 2001 to June 2017 |

Figure 9 and Tab. 6 compares the SWE automatic measurements at location 16 (Fig. 2) with the manual measured in the
5  main snow pit field (panel a) and the three locations for manual SWE measurements (panel b). The statistics are calculated



over the 2001-2016 period. It must be underlined that the SWE automatic sensor is calibrated using the manual measurements at snow pit fields south and north.

Figure 9 and Tab. 6 show that the mean difference between the automatic and the manual measurements in the main snow pit field reaches -17 kg m$^{-2}$ with RMSD of almost 25 kg m$^{-2}$. The comparison between the three locations of manual measurements displays RMSD reaching 25 kg m$^{-2}$. This value is consistent with the spatial variability of snow depth and can probably be used as an estimate of the uncertainty associated to the SWE dataset both due to measurements errors and spatial variability.

### 3.4 Soil temperature

**Table 7.** Statistics of the comparisons between the different soil temperature measurements represented in Fig. 10.

| Sensors | Depth (cm) | Number of dates | Deviation (K) | RMSD (K) | Period |
|---|---|---|---|---|---|
| s2_loc23 - s1_loc23 | 10 | 15084 | 0.034 | 0.110 | Dec. 2015 to June 2017 |
| s3_loc23 - s1_loc23 | 10 | 15084 | -0.094 | 0.244 | Dec. 2015 to June 2017 |
| s3_loc23 - s2_loc23 | 10 | 15084 | 0.128 | 0.182 | Dec. 2015 to June 2017 |
| loc_23 - loc_24 | 10 | 11396 | -0.108 | 0.415 | Dec. 2015 to June 2017 (snow season) |
| loc_23 - loc_24 | 10 | 3688 | -1.059 | 1.100 | Dec. 2015 to June 2017 (summer) |
| s2_loc23 - s1_loc23 | 20 | 15084 | 0.093 | 0.118 | Dec. 2015 to June 2017 |
| loc_23 - loc_24 | 20 | 11396 | -0.224 | 0.390 | Dec. 2015 to June 2017 (snow season) |
| loc_23 - loc_24 | 20 | 3688 | -0.943 | 0.961 | Dec. 2015 to June 2017 (summer) |

Figure 10 and Tab. 7 compare the different soil temperature measurements at 10 and 20 cm depths for locations 23 and 24 (Fig. 2). The first column display the statistics of the different temperature probes located index 23 and spaces by roughly 10 cm. It indicates that the RMSD between the probes is lower than 0.25 K. The second column compares location 24 (old sensors) and 23 (new sensors) for two periods : summer (20 June to 10 October) and snow season (11 October to 19 June). During the snow season, the two locations show a small mean deviation of -0.11K and an RMSD of 0.42 K while during summer the mean deviation is roughly -1.06 K leading to RMSD of 1.10 K. Note that these two locations are spaced by a few meters only (see Fig. 2).

From these observations, a lower bound of the uncertainty of the soil temperature measurements (spatial variability and measurements errors) is roughly 1.10 K during summer, roughly 0.42 K during the snow season and a little higher than 0.5 K averaged over the whole year.

## 4 Data use

### 4.1 Temperature, snow depth and precipitation since 1960

Fig. 11 displays the evolution of mean snow depth, air temperature and total precipitation from Dec. 1$^{st}$ to April 30$^{th}$ of each snow season for the whole period of the dataset (Dec. 1960 - April 2017). This figure shows an example of a direct use of the dataset to study the past evolution of winter conditions at Col de Porte. It demonstrates that the mean snow depth reduction





**Figure 10.** Comparison between the different soil temperature measurements at 10 cm (panels a and b) and 20 cm (panels c and d) depths. Panels a and c compare the new sensors (3 probes roughly 10 cm away of each other at location 23 at -10 cm and 2 probes roughly 10 cm away of each other at -20 cm). Panels b and c compares the average values of the new sensors (location 23) to the old ones (location 24). Statistics are calculated from December 2015 to July 2017. Summer (panels b and d, in red) corresponds to the period between 20 June 2016 and 10 October 2016 and 20 June 2017 to 31 July 2017. The rest of the dates corresponds to snow season (panels b and d, in blue). Numerical values are provided in Tab. 7.

between 1960-1990 and 1990-2017 is 39 cm, while the air temperature has increased of 0.90 °C over the same period and the total precipitation does not exhibit a significant trend. This indicates that at this site, the reduction of the snow cover is mainly



due to the increase in temperature and its consequences (e.g. higher snow/rain limit when precipitation and higer melt rates). This long time serie contributes to placing long term climate change impact studies on mountain snow conditions in context of past changes (Verfaillie et al., 2018).

### 4.2 Snow model evaluation

5 This dataset has been widely used to drive and evaluate snow models (e.g. Essery et al., 2013; Magnusson et al., 2015; Decharme et al., 2016; Lafaysse et al., 2017; Piazzi et al., 2018). A list of the studies using CDP dataset is available at http://www.umr-cnrm.fr/spip.php?article533. It has also been included in the Earth System Model - Snow Model Intercomparison Project (ESM-SnowMIP).

### 5 Data availability

10 The database (doi:10.17178/CRYOBSCLIM.CDP.2018) presented and described in this article is available for download at http://dx.doi.org/10.17178/CRYOBSCLIM.CDP.2018. Tab. 8 provides the links to the different datasets.

**Table 8.** Link to the dataset repository

| Dataset | Period | Format | Repository |
|---|---|---|---|
| Solar Mask | July 1998 and June 2018 | csv | http://dx.doi.org/10.17178/CRYOBSCLIM.CDP.2018.SolarMask |
| Soil properties | 29 September 2008 and October 2nd 2012 | csv | http://dx.doi.org/10.17178/CRYOBSCLIM.CDP.2018.Soil |
| Hourly in situ meteorological data | August 1st 1993 to 31 July 2017 | netCDF | http://dx.doi.org/10.17178/CRYOBSCLIM.CDP.2018.MetInsitu |
| Hourly SAFRAN meteorological data | August 1st 1960 to 31 July 2017 | netCDF | http://dx.doi.org/10.17178/CRYOBSCLIM.CDP.2018.MetSafran |
| Daily snow and meteorological data | August 1st 1960 to 31 July 2017 | netCDF | http://dx.doi.org/10.17178/CRYOBSCLIM.CDP.2018.MetSnowDaily |
| Hourly snow data | August 1st 1960 to 31 July 2017 | netCDF | http://dx.doi.org/10.17178/CRYOBSCLIM.CDP.2018.HourlySnow |
| Snow profiles | September 1993 to April 2015 | caaml | http://dx.doi.org/10.17178/CRYOBSCLIM.CDP.2018.SnowProfile |

*Author contributions.* Y.L. and J.-M. P. endorse the responsability of the experimental site and of the instruments. M.D. lead the consolidation of the data set and wrote this manuscript together with all co-authors. E. L. and J.-M. P. ensure the proper working of the instruments.

*Acknowledgements.* Many thanks are expressed to people of CNRM/CEN who have assisted in the collection, collation and archive of
15 this unique dataset. We thank in particular É. Pougatch, Y. Danielou, J.-L. Dumas for their crucial work on the database. Generating and distributing this dataset directly benefitted from the LabEX OSUG@2020 (ANR10 LABX56). The authors are also thankful to EDF/DTG, ONF, SOERE CryObsClim, GCW and the IDEX Univ. Grenoble Alpes Cross Disciplinary Project Trajectories. The authors also thanks Arnaud Foulquier (LECA) for his help with soil and vegetation properties measurements and Laurent Bourges (OSUG) for the establishment of the data repository and the allocation of dedicated dois.





**Figure 11.** Evolution of mean snow depth, air temperature and total precipitation over 1960-2017. The mean and total values are calculated over the period Dec. 1st to April 30th of each snow season. The black lines are the 15 years moving means.



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
