# Peer review of "years (1960-2017) of snow and meteorological observations from a mid-altitude mountain site (Col de Porte, France, 1325 m alt.)"

_Earth System Science Data, 2018_

## Referee Comment (RC1) · R. L. H. Essery (Referee) · 7 Sep 2018

The availability of such a long, comprehensive, well-maintained and well-documented dataset is important for snow modelling, and I am keen to see this paper published. I have only a few questions:

page 2, line 14

It might be appropriate here to mention the important contribution of Col de Porte data to SnowMIP (doi:10.3189/172756404781814825) and ESM-SnowMIP (https://doi.org/10.5194/gmd-2018-153).

[Figure]

page 2, line 20

The underlined text only covers instrument types and periods in tables. I do not think that the underlining is necessary.

Section 2.1

It would be interesting to know a bit more about how the elevations and $p_{occ}$ are measured.

Figure 2

Some symbol indicating the direction of view would be preferable to the emoticons for camera locations.

Figure 3

It is a little deceptive having the centre of the masks at 60 degrees elevation rather than 90 degrees.

page 8, line 3

Is "available in that study" intended, i.e. Morin et al. (2012)?

page 8, line 23

Equation (2) will simplify a bit; is there a reason why this is not done? Does it always have one and only one solution for $\eta$ in reasonable ranges of epsilon and $e_{air}$?

page 9, line 7

The signs of the corrections are the wrong way round, aren't they?

Figure 6

The red temperature line is impossible to see on the red hardness bars.

page 14, line6

I think that this means "mitigating the impact of undercatch", not the impact of the undercatch correction.

page 16, line 22

"mask measured in 2018"

page 16, line 26

The location of the total and diffuse radiation measurements is given as 2 here and 5 in the Figure 7 caption.

Figure 9

The (a) legend should be "Pit – reference"

page 20, line 9

References to the first and second columns of Table 7 do not seem to make sense.

The writing is always clear, but I have sent the authors some minor corrections directly.

---

## Referee Comment (RC2) · Anonymous Referee #2 · 21 Sep 2018

Long term meteorological data are crucial for climate variability studies. It is of the utmost importance that these data are carefully collected, QCed, and stored, as well as fully documented (metadata). In that perspective, this paper presents a perfect example of the above, and it is important to support and promote such datasets to be available for the community.

I recommend to accept this paper, with minor corrections, which are listed here:

- page 1, line 6: Unit is missing after 0.21.

- p1, l7: .... that can mainly be .... for snow water equivalent

[Figure]

- p1, l9: Reduction of 39 cm in the mean snow depth, but what does it represent in %age of the mean total snow depth?

- p1, l17: required to run and evaluate

- p2, l15: (i) to extend

- p2, l16: (ii) to provide .... (iii) to provide

- p2, l20: I do not understand this sentence, and could not find underlined text in the paper => remove (?)

- p8, l7: Explain why the in-situ data are missing during Summer between 2011 and 2015

- p9, l5-10: The process of the correction is not "clear and clean" to me. Some information are missing: what is the Impact of a 10 W/m2 shift (or error) on the snow pack model? how does the final curve in Fig. 5 Looks like? is monthly average enough to assess the quality of the data (variation can be much higher on an hourly basis).

- Table 2: CGR4 (and not CRG4)

- p10, l3: What are the relevant sources of data (list)?

p10, l6-7: If wind data used to correct for undercatch is different, then the correction factors must also be different, right?

- p14, l4: .... not corrected for undercatch, in contrary ....

- p14, l5: .... by the same sensor used for rain and snow datasets

- p14, l6: Explain why the undercatch (and not the undercatch correction?) is mitigated when using the PG2000

- Table 5: For clarity, it should be moved in section 3.2 (where it is referenced)

- p20, l13: Explain why the mean deviation is higher in summer (shading, surface
properties?)

- I miss a short conclusion at the end of the paper. Few words summarizing the work and the dataset available.

---

## Referee Comment (RC3) · C. Fierz (Referee) · 10 Oct 2018

General comments The authors present an important addition to the dataset described by Morin et al. in 2012. Not only is the hourly 2012 data set prolonged to 2017, but they also compiled all data available at Col de Porte in a daily dataset starting in 1960. The data are conveniently described and presented clearly and concise-ly without loosing on clarity, except for soil temperatures (see Section 3.4 in annotated manuscript). The estimation of uncertainties of the various measurements is an important asset of the paper as these are rarely given for other data sets. In conclusion, the paper is a welcomed contribution to long term, well described data sets for snow studies. Thus I

recommend to accept the paper provided the authors address the points raised above and below as well as in the annotated version of their manuscript.

Specific comments • Regarding uncertainties on the water equivalent of snow cover (SWE), I have two questions: 1) Would it not be more interesting to compare the bulk density of the snow cover given by SWE/HS (where HS is snow depth) rather than SWE itself ? Indeed, I assume that the spatial varia-bility of SWE is mostly given by the spatial variability of HS while measurement errors of the snow mass may dominate the uncertainty on bulk density. I am aware that the cosmic-ray sensor does not fully fit this approach, but I assume from Figure 2 that snow depth is quite well known at its loca-tion. 2) If comparing SWE, I would prefer to see the differences expressed as a percentage of SWEref. But at least the mean maximum SWE should be indicated. • Regarding Fig., Tab., etc, I would suggest to use the same style throughout the paper. Myself I would tend to use Figure, Table, Section, etc. • The reader often needs to navigate back to Figure 2 (and other Tables). I hope this can be made eas-ier by linking any Location N to Figure 2 as well as referrals to tables to their respective Entries. That way the multiple repetition of '(Figure 2)' could be avoided in the text. Anyway, whatever you decide, make it consistent throughout the paper (currently not the case). Moreover, I hope the final layout will also help with that respect. Currently, some Figures and Ta-bles seem badly misplaced.

Please also note the supplement to this comment:
https://www.earth-syst-sci-data-discuss.net/essd-2018-84/essd-2018-84-RC3-supplement.pdf

**Supplement:**

[revised manuscript text omitted]

---

## Author Comment (AC1) · 7 Nov 2018

**Author response to comments on «57 years (1960–2017) of snow and meteorological observations from a mid-altitude mountain site (Col de Porte, France, 1325 m alt.)» by Yves Lejeune et al.**

The authors are grateful to the 3 referees for the time they devoted to review the manuscript and for their useful comments. Below is a point by point response to each comments. Authors responses are in blue and changes in the manuscript are enlightened in **bold**.

**Comment by R. L. H. Essery (Referee #1)**

**Additional changes in the manuscript :**
The dataset of snow vertical profiles has been extended to March 2018 (April 2015 in the first version of the manuscript) and these profiles are now provided in caaml v6 format (caaml v5 in the first version).

The availability of such a long, comprehensive, well-maintained and well-documented dataset is important for snow modelling, and I am keen to see this paper published. I have only a few questions:

The authors are very grateful to R. Essery for his positive, careful and useful review of the manuscript. Below is a point by point response to each comments. Authors responses are in blue and changes in the manuscript are enlightened in **bold**.

page 2, line 14
It might be appropriate here to mention the important contribution of Col de Porte data to SnowMIP (doi:10.3189/172756404781814825) and ESM-SnowMIP (https://doi.org/10.5194/gmd-2018-153).

Thanks for the references that have been added in the manuscript which now reads : « ... (ONERC). **The CDP dataset has been used as driving and evaluation data in several snow model intercomparison projects : SnowMIP (Etchevers et al., 2004) and ESM-SnowMIP (Krinner et al., 2018).** »
The explicit reference to ESM-SnowMIP page 22 line 7 has been consequently removed.

page 2, line 20
The underlined text only covers instrument types and periods in tables. I do not think that the underlining is necessary.
The « track changes » with respect to the former description of the dataset (Morin et al., 2012) was requested so that the paper was referenced in the Living data process of ESSD. Since the two referees found it unnecessary and misleading, the underlining has been removed in the revised manuscript.

Section 2.1
It would be interesting to know a bit more about how the elevations and p_occ are measured.

The 1998 masks were measured using a theodolite and the 2018 ones using a compass and a clinometer. Values of p_occ were visually estimated. This information have been added in the text as follows :

"Table \ref{tab:driving}) with 5$^\circ$ resolution in azimuth for two dates: July 1998 (**using a theodolite**) and June 2018 (**using a compass and a clinometer**). Masks are provided as a .csv file (\href{http://dx.doi.org/10.17178/CRYOBSCLIM.CDP.2018.SolarMask} {doi:10.17187/CRYOBSCLIM.CDP.2018.SolarMask}), they contain 3 values for each azimuth that correspond to: lower elevation, upper elevation and occultation percentage ($p_{occ}$, **visually estimated**) defined as follows (Fig. \ref{fig:mask})."

Figure 2 Some symbol indicating the direction of view would be preferable to the emoticons for camera locations.
The 3 cameras are hemispherical, this information was added in Figure 2 captions and the emoticons were replaced by dark blue asterisks. The caption now reads :

"Schematic view of the experimental sites with sensor locations. The sensors indicated in yellow are for meteorological variables. The sensors indicated in red are not used anymore as of 2018, and those in blue correspond to snow measurements. Areas 23 and 24 correspond to soil temperature and humidity measurements. The correspondance between numbering and sensors is indicated in Tables \ref{tab:driving}, \ref{tab:eval} and \ref{tab:quot}. **The three dark blue asterisks correspond to the three hemispherical** Webcam locations."

New Figure 2 is shown below :

[Figure]

Figure 3 It is a little deceptive having the centre of the masks at 60 degrees elevation rather than

90 degrees.

Figure 3 has been modified, and the center of the mask is now 90° as shown below.

[Figure]

page 8, line 3 Is "available in that study" intended, i.e. Morin et al. (2012)?

Yes, we changed the sentence accordingly.

« The meteorological hourly dataset over 1993-2017 is an extension of the meteorological dataset provided in \citet{morin2012b} **in which an extensive description of the dataset is available.**»

page 8, line 23 Equation (2) will simplify a bit; is there a reason why this is not done? Does it always have one and only one solution for η in reasonable ranges of epsilon and e air ?

Thanks for this comment. There were several mistakes in Equations (1) and (2). The changes have been performed as follows :

"The first step of the procedure is to compute a cloudiness value, $\eta$ (no unit, between 0 for clear sky and 1 for fully overcast) from measured air temperature $T_{air}$ (K), longwave radiation $LW_{down}$ (W m$^{-2}$) and specific humidity using Eqs. (\ref{eq:nebul1}) and (\ref{eq:nebul2}) from \citet{Berliand1952, **etchevers2000**}.
\begin{equation}
LW_{down}=**1.05** \varepsilon \sigma T_{air}^4
\label{eq:nebul1}
\end{equation}

**\begin{equation}**
**\varepsilon=0.58+0.9k(\eta)+0.06\sqrt{e_{air}}(1-k(\eta))**
**\label{eq:nebul2}**
**\end{equation}**

**\begin{equation}**
**k(\eta)=(0.09+0.2\eta)\eta^2**
**\label{eq:nebul3}**
**\end{equation}**

where $\sigma$ is the Stefan-Boltzman constant, and $e_{air}$ is the   water vapour partial pressure calculated from measured $T_{air}$ and relative humidity, expressed in hPa. **The correction factor $1.05$ in Eq. (\ref{eq:nebul1}) accounts for the additional longwave radiation**

**that is reaching the sensor due to the presence of surrounding trees. Eq. (\ref{eq:nebul2}) solution does not necessarily range between 0 and 1, $\eta$ must be bounded between 0 and 1 when solving the equation.**"

page 9, line 7 The signs of the corrections are the wrong way round, aren't they?
Yes, thanks for noticing. This had been corrected.

Figure 6 The red temperature line is impossible to see on the red hardness bars.
Yes, the color of the temperature profile has been changed to black as shown below.

[Figure]

page 14, line6 I think that this means "mitigating the impact of undercatch", not the impact of the undercatch correction.
Yes, thanks for noticing. It has been corrected.

page 16, line 22 "mask measured in 2018"
Yes, thanks for noticing. It has been corrected.

page 16, line 26 The location of the total and diffuse radiation measurements is given as 2 here and 5 in the Figure 7 caption.
Thanks for noticing, the correct location is 5, it has been corrected in the text.

Figure 9 The (a) legend should be "Pit – reference"
Yes, this has been changed as shown below.

[Figure]

page 20, line 9 References to the first and second columns of Table 7 do not seem to make sense.
Yes. It has been modified. It is the first and second column of figure 10 not table 7. The sentences have been changed.
« The **left panels in Fig. \ref{fig:soil} display** the statistics of the different temperature probes at location 23 and spaced by roughly 10 cm. It indicates that the RMSD between the probes is lower than 0.25 K **(Tab. \ref{tab:soil})**. The **right panels in Fig. \ref{fig:soil} compare** locations 24 (old sensors) and 23 (new sensors) for two periods : summer (20 June to 10 October) and snow season (11 October to 19 June). During the snow season, the two locations show a small mean deviation of -0.11K and an RMSD of 0.42 K, while during summer the mean deviation is roughly -1.06 K leading to RMSD of 1.10 K **(Tab. \ref{tab:soil})**. Note that these two locations are spaced by only a few meters (see Fig. \ref{fig:scheme}). »

The writing is always clear, but I have sent the authors some minor corrections directly.
Thanks a lot for taking time to do the corrections, all corrections have been accounted for and are enlightened in the track-change pdf of the manuscript submitted along with the revised manuscript.

---

## Author Comment (AC2) · 7 Nov 2018

**Comments by Anonymous Referee #2**

Long term meteorological data are crucial for climate variability studies. It is of the utmost importance that these data are carefully collected, QCed, and stored, as well as fully documented (metadata). In that perspective, this paper presents a perfect example of the above, and it is important to support and promote such datasets to be available for the community.

The authors are very grateful to the anonymous reviewer for his/her positive, careful and useful review of the manuscript. A point by point response to each comment is provided in blue below. Changes in the manuscript are enlightened in **bold**.

**Additional changes in the manuscript :**
The dataset of snow vertical profiles has been extended to March 2018 (April 2015 in the first version of the manuscript) and these profiles are now provided in caaml v6 format (caaml v5 in the first version).

I recommend to accept this paper, with minor corrections, which are listed here:

- page 1, line 6: Unit is missing after 0.21.
It's ratio of irradiance so no unit. This has been added in the text.

- p1, l7: .... that can mainly be .... for snow water equivalent
Thanks, the sentence has been changed to :
« The estimated root mean squared deviation, **which mainly represents** spatial variability, is $\pm$ 10 cm for snow depth, $\pm$ 25 kg m$^{-2}$ for snow water **equivalent** and $\pm$ 1 K for soil temperature ($\pm$ 0.4 K during the snow season). »

- p1, l9: Reduction of 39 cm in the mean snow depth, but what does it represent in %age of the mean total snow depth?
39 cm represents 40% of the mean total snow depth for the period 1960-1990. This information was added in the sentence as follow :
"The daily dataset can be used to quantify the effect of climate change at this site with a reduction of the mean snow depth (Dec. 1$^{\rm{st}}$ to April 30$^{\rm{th}}$) of 39 cm from 1960-1990 to 1990-2017 **(40 $\%$ of the mean snow depth for 1960-1990)** and an increase in temperature of + 0.90 K for the same periods."
This information has also been added page 22 line 6 :
"It demonstrates that the mean snow depth reduction between 1960-1990 and 1990-2017 is 39 cm **(40 $\%$ of the mean snow depth for 1960-1990)**, while the air temperature has increased by 0.90 $^{\circ}$C over the same period and the total precipitation does not exhibit a significant trend."

- p1, l17: required to run and evaluate
The sentence has been changed accordingly.

- p2, l15: (i) to extend
Ok, this has been changed.

- p2, l16: (ii) to provide .... (iii) to provide
Ok, this has been changed.

- p2, l20: I do not understand this sentence, and could not find underlined text in the paper => remove (?)
The « track changes » with respect to the former description of the dataset (Morin et al., 2012) was requested so that the paper was referenced in the Living data process of ESSD. Since the two referees found it unnecessary and misleading, the underlining has been removed in the revised manuscript.

- p8, l7: Explain why the in-situ data are missing during Summer between 2011 and 2015
Historically, Col de Porte was dedicated to snow studies. Consequently, most of the sensors were removed and calibrated during summer. This is the reason why summer data are missing from 1993 to 2015 (see fig. 4 in Morin et al., 2014). In 2015, we decided to maintain the measurements during the whole year.
The text has consequently been modified as follows :
"The partitioning of the dataset between \textit{in-situ} data and the output of the meteorological analysis and downscaling tool SAFRAN \citep{durand1999, **durand2009**} is the same as in Fig. 4 of \citet{morin2012b}. For years 2011 to 2015, \textit{in-situ} data are restricted to the period 20 October of one year to 10 June of the next year. **Summer \textit{in situ} data are thus missing (calibration of the sensors during summer) from 1993 to 2015**. Starting on 10 June 2015, all data are \textit{in-situ} year-round except for very short periods with observation issues."

- p9, l5-10: The process of the correction is not "clear and clean" to me. Some information are missing: what is the Impact of a 10 W/m2 shift (or error) on the snowpack model? how does the final curve in Fig. 5 Looks like? is monthly average enough to assess the quality of the data (variation can be much higher on an hourly basis).

The impact of the longwave radiation correction on the simulations continuously increases along each winter season and becomes maximum during the melting period in spring. With the Crocus snowpack model, the yearly maximum difference in terms of snow depth ranges between 30 and 60 cm depending on the year, and between 150 and 300 kg/m² in terms of water equivalent of snow cover. In winter, a 2 K difference in surface temperature is common, with some much higher values. At the end of the season, the shift in the date of total melt out ranges between 5 and 10 days.

The curve in Fig. 5 corresponding to the final observation product is simply a plus or minus 10 W/m² shift during the two periods when the correction was considered necessary, as now shown by the green curve. This allows removing the breaks although there is still a noise between SAFRAN and observations. The bias identification was obtained from monthly comparisons between Col de Porte measurements and SAFRAN simulations for the Chartreuse massif at the same elevation. Only the significant temporal breaks in this difference can reasonably be attributed to instrumental issues. At shorter time steps (not shown here), and especially at an hourly time step, the differences between local observations and massif-scale simulations exhibit fluctuations, which are most likely due to local topographic effects, potential discrepancies between the local cloudiness and the simulated massif-scale cloudiness, etc. SAFRAN is the only other available reference because there was only one sensor. Therefore, it is unfortunately not possible to

investigate with more temporal refinement the instrumental biases. Note also that the impact on snowpack simulations is mostly sensitive to systematic biases.

The text and figures were consequently modified as follows :
"Based on the hypothesis that the newest sensor **can be used as** a reference **because** it was fully calibrated at the **Physikalisch-Meteorologisches Observatorium (Davos, Switzerland)** outside and inside with a blackbody, the dataset was corrected as follow : -10 W m$^{-2}$ from 1993 to November 2010 and +10 W m$^{-2}$ from November 2010 to November 2015. **Since SAFRAN is the only available reference and does not account for local conditions, e.g. cloudiness, due to its coarse spatial resolution, it is unfortunately not possible currently to investigate with more temporal refinement this instrumental bias.** This correction, although **panning** the uncertainty values provided by the manufacturer, is of large significance for snowpack modelling considering the high sensitivity of the snowpack to **processes governed by** this variable (e.g. \citealp{raleigh2014, sauter2015, queno2017}). **Using the Crocus snowpack model with or without the corrections leads to a shift in the melt-out date ranging between 5 and 10 days (not shown).** "

[Figure]

- Table 2: CGR4 (and not CRG4)
Thanks, it has been modified.

- p10, l3: What are the relevant sources of data (list)?
This information has been added in the text :
"Precipitation data are manually partitioned between liquid and solid phase using all relevant sources of data at the site, **namely snow depth, surface albedo, surface and air temperatures and differences between heated and non-heated rain gauges (locs. 1 and 9)**."

p10, l6-7: If wind data used to correct for undercatch is different, then the correction factors must also be different, right?
Yes, we agree and the sentence was misleading. Locations 15 (used before 2013) and 18 (used after 2013) are very close. Thus in this case, we believe it is reasonable to assume no difference in the correction factors.
The text has been updated as follows:
"From 2013 to 2017, the wind measurement used for the correction **was the one** placed at location 18 (Fig. \ref{fig:scheme}) **instead of location 15, since the ultrasonic sensor at location 18**

**is more accurate than the wind sensor at location 15. Note that locations 15 and 18 are very close, i.e. a few meters, so that the wind speed values are not significantly different between the two locations.** "

- p14, l4: …. not corrected for undercatch, in contrary ….
Ok, it has been corrected.

- p14, l5: …. by the same sensor used for rain and snow datasets
Ok, it has been corrected.

- p14, l6: Explain why the undercatch (and not the undercatch correction?) is mitigated when using the PG2000
The undercatch is inversely proportional to the collecting area. Since the PG2000 collecting area (2000 cm²) is 10 times larger than the GEONOR collecting area (200 cm²) , the undercatch is thus less important for the PG2000. The comparison in case of snowfalls between the PG2000 (not corrected for undercatch) and the GEONOR (corrected for undercatch) showed a very good agreement.
The information was added in the text as follows:
"The total precipitation dataset is measured with the PG2000 sensor, **for which the undercatch plays a minor role compared to the GEONOR due to the 10 times larger collecting surface area ( Table 2)."**

- Table 5: For clarity, it should be moved in section 3.2 (where it is referenced)
Ok, Table 5 have been moved to section 3.2.

- p20, l13: Explain why the mean deviation is higher in summer (shading, surface properties?)
An explanation has been added in the paragraph as follows:
**«The temperature difference between the two sensors may be attributed to differences in soil properties, local topography and shading. The larger differences in summer may be due to (i) larger heterogeneity in soil wetness and (ii) the absence of the snow cover that spatially tempers the surface temperature signal in winter.»**

- I miss a short conclusion at the end of the paper. Few words summarizing the work
and the dataset available.
A short conclusion has been added to the paper and reads:
"\conclusions
**This paper describes and provides access to the daily snow and meteorological dataset measured at the Col de Porte site, 1325 m a.s.l, Chartreuse, France for the period 1960-2017. The hourly dataset of snow and meteorological observations for the period 1993-2017 is made available along with weekly snow profiles from September 1993 to April 2015, soil properties and solar radiation masks. Based on measurements at several locations within the measurement field, we estimated the uncertainties and spatial variability of : the ratio between solar diffuse and total irradiance, snow depth, snow water equivalent and soil temperature. The data are placed on the repository of the** Observatoire des Sciences de l\textquotesingle Univers de Grenoble (OSUG) datacenter :
\href{http://doi.osug.fr/public/CRYOBSCLIM_CDP/CRYOBSCLIM.CDP.2018.html}
{http://dx.doi.org/10.17178/CRYOBSCLIM.CDP.2018}. "

---

## Author Comment (AC3) · 7 Nov 2018

**Comments by C. Fierz**

**General comments**

The authors present an important addition to the dataset described by Morin et al. in 2012. Not only is the hourly 2012 data set prolonged to 2017, but they also compiled all data available at Col de Porte in a daily dataset starting in 1960. The data are conveniently described and presented clearly and concisely without loosing on clarity, except for soil temperatures (see Section 3.4 in annotated manuscript). The estimation of uncertainties of the various measurements is an important asset of the paper as these are rarely given for other data sets. In conclusion, the paper is a welcomed contribution to long term, well described data sets for snow studies. Thus I recommend to accept the paper provided the authors address the points raised above and below as well as in the annotated version of their manuscript.

The authors are very grateful to Charles Fierz for his careful and useful review of the manuscript. A point by point response to each comment is provided in blue below. Changes in the manuscript are enlightened in **bold**.

**Additional changes in the manuscript :**
The dataset of snow vertical profiles has been extended to March 2018 (April 2015 in the first version of the manuscript) and these profiles are now provided in caaml v6 format (caaml v5 in the first version).

**Specific comments**

Regarding uncertainties on the water equivalent of snow cover (SWE), I have two questions:

1) Would it not be more interesting to compare the bulk density of the snow cover given by SWE/HS (where HS is snow depth) rather than SWE itself ? Indeed, I assume that the spatial variability of SWE is mostly given by the spatial variability of HS while measurement errors of the snow mass may dominate the uncertainty on bulk density. I am aware that the cosmic-ray sensor does not fully fit this approach, but I assume from Figure 2 that snow depth is quite well known at its location.

Yes, we agree that this would be interesting, especially regarding discussion on the spatial variability of the bulk density and its density such as in Sturm et al., 2010. Unfortunately, no snow depth sensor is located close enough to the reference SWE sensor (location 16 in Fig. 2). We are planning, in a near future, to add a snow depth sensor at this location. The snowpits data, which combine both SWE and HS measurements can be used to investigate the spatial variability of bulk density. Since the reference SWE sensor is not combined with snow depth measurement we believe that it is more relevant in that case to stick to the estimation of SWE variability and uncertainties.

*Sturm, M., B. Taras, G.E. Liston, C. Derksen, T. Jonas, and J. Lea, 2010:* Estimating Snow Water Equivalent Using Snow Depth Data and Climate Classes. *J. Hydrometeor., 11, 1380–1394,* https://doi.org/10.1175/2010JHM1202.1

2) If comparing SWE, I would prefer to see the differences expressed as a percentage of SWEref. But at least the mean maximum SWE should be indicated.
Translating Fig. 9 in percent of SWE ref would result in the following figure :

[Figure]

We believe that the representation with the absolute values give less emphasis on the outliers and thus we prefer to keep the analysis in absolute values in the revised manuscript. However, as suggested we added in the text the mean value of peak SWE and added the percentage in mean peak SWE in the description of the figure.

"Figure \ref{fig:swe} and Table \ref{tab:swe} compare the SWE automatic measurements at \hyperref[fig:scheme]{location 16} with the manual measurements from the main snow pit field (panel a) and the three locations for manual SWE measurements (panel b). The statistics are calculated over the 2001-2017 period. It must be underlined that the automatic SWE sensor is calibrated using the manual measurements at snow pit fields south and north. **The average of the annual maximum value of SWE$_{\rm{ref}}$ during this period is 389 $\pm$ 104 kg~m$^{-2}$ .** Figure \ref{fig:swe} and Table \ref{tab:swe} show that the mean difference between the automatic and manual measurements in the main snow pit field reaches -17 kg\,m$^{-2}$ with RMSD of almost 25 kg\,m$^{-2}$. The comparison between the three locations of manual measurements displays RMSD reaching 25 kg\,m$^{-2}$, **i.e. 8.6 $\%$ of average peak SWE values.** This value is consistent with the spatial variability of snow depth and can probably be used as an estimate of the uncertainty associated with the SWE dataset both due to measurement errors and spatial variability. "

Regarding Fig., Tab., etc, I would suggest to use the same style throughout the paper. Myself I would tend to use Figure, Table, Section, etc.

According to ESSD website (https://www.earth-system-science-data.net/for_authors/manuscript_preparation.html),

"The abbreviation "Fig." should be used when it appears in running text and should be followed by a number unless it comes at the beginning of a sentence, e.g.: "The results are depicted in Fig. 5. Figure 9 reveals that..."."
"Please note that the word "Table" is never abbreviated and should be capitalized when followed by a number (e.g. Table 4)."
**"Equations**: They should be referred to by the abbreviation "Eq." and the respective number in parentheses, e.g. "Eq. (14)". However, when the reference comes at the beginning of a sentence, the unabbreviated word "Equation" should be used, e.g.: "Equation (14) is very important for the results; however, Eq. (15) makes it clear that...""
"The abbreviation "Sect." should be used when it appears in running text and should be followed by a number unless it comes at the beginning of a sentence."

These guidelines have been taken into accounted in the new version of the manuscript (also pointed out by R. Essery). We also note that the Copernicus Editorial team will prepare the manuscript for final publication and enforce style issues that may have escaped our attention.

The reader often needs to navigate back to Figure 2 (and other Tables). I hope this can be made easier by linking any Location N to Figure 2 as well as referrals to tables to their respective Entries. That way the multiple repetition of '(Figure 2)' could be avoided in the text. Anyway, whatever you decide, make it consistent throughout the paper (currently not the case).

Tables references were already linked to their respective entries.
We have now linked every location to Figure 2 in .pdf.
This was also added in the legend of Figure 2 that now reads :
"...The correspondence between numbering and sensors is indicated in Tables \ref{tab:driving}, \ref{tab:eval} and \ref{tab:quot}. **For the sake of clarity, when a location is cited in the text, the reference to Fig. \ref{fig:scheme} is omitted and the location is directly linked to the figure or the corresponding table.** ..."

Moreover, I hope the final layout will also help with that respect. Currently, some Figures and Tables seem badly misplaced.

This formatting issue will be dealt with during the final editing process, upon acceptance of the manuscript, in connection with the publishing staff. The current locations of figures and tables have been also checked and updated (see also response to comment from referee #2).

Please also note the supplement to this comment: https://www.earth-syst-sci-data-discuss.net/essd-2018-84/essd-2018-84-RC3-supplement.pdf

Each comment and annotation in the supplement have been taken into account (please see the track-changes version of the manuscript submitted along with the revised version of the manuscript).

Regarding the comment of the HTN diff, the 80 cm difference values was a mistake in the dataset. Thanks for spotting it ! It has been corrected and Fig. 8 and the dataset has been updated consequently.